# Thrombocytopathies: Not Just Aggregation Defects—The Clinical Relevance of Procoagulant Platelets

**DOI:** 10.3390/jcm10050894

**Published:** 2021-02-24

**Authors:** Alessandro Aliotta, Debora Bertaggia Calderara, Maxime G. Zermatten, Matteo Marchetti, Lorenzo Alberio

**Affiliations:** 1Hemostasis and Platelet Research Laboratory, Division of Hematology and Central Hematology Laboratory, Lausanne University Hospital (CHUV) and University of Lausanne (UNIL), CH-1010 Lausanne, Switzerland; Alessandro.Aliotta@chuv.ch (A.A.); Debora.Bertaggia-Calderara@chuv.ch (D.B.C.); Maxime.Zermatten@chuv.ch (M.G.Z.); matteo.marchetti@ghol.ch (M.M.); 2Service de Médecine Interne, Hôpital de Nyon, CH-1260 Nyon, Switzerland

**Keywords:** thrombocytopathy, platelet disorders, procoagulant platelets, activation endpoints

## Abstract

Platelets are active key players in haemostasis. Qualitative platelet dysfunctions result in thrombocytopathies variously characterized by defects of their adhesive and procoagulant activation endpoints. In this review, we summarize the traditional platelet defects in adhesion, secretion, and aggregation. In addition, we review the current knowledge about procoagulant platelets, focusing on their role in bleeding or thrombotic pathologies and their pharmaceutical modulation. Procoagulant activity is an important feature of platelet activation, which should be specifically evaluated during the investigation of a suspected thrombocytopathy.

## 1. Introduction

Platelets or thrombocytes are small (2–5 µm) discoid anucleated cells produced by megakaryocytes. They are released in the blood stream where they circulate for 7–10 days to be eventually cleared by the spleen and the liver [1]. Platelets are responsible for maintaining the integrity of the vascular system, are active key players of primary haemostasis and enhance coagulation. Consequently, platelet disorders cause defective clot formation that may induce a bleeding or thrombotic diathesis.

Platelet disorders can be either inherited or acquired and are characterized by (i) quantitative defects, with an abnormal number of circulating platelets, either high (thrombocytosis) or low (thrombocytopenia); and/or (ii) qualitative platelet dysfunctions (thrombocytopathies) [2].

Thrombocytopathies may be induced either by extrinsic (e.g., systemic disease or medication) or by intrinsic factors [3,4]. In this review, we summarize intrinsic platelet anomalies resulting in defects of the various traditional activation endpoints, such as adhesion and aggregation (See Section 2), and we offer an in-depth and complete overview of the accumulating evidence for the physiological and clinical role of procoagulant platelets as an alternative, increasingly recognized critical endpoint of platelet function (see Section 3 and Section 4).

## 2. Platelet Activation End-Points and Related Defects

At the site of vascular damage, platelets interact with exposed adhesive agonists such as von Willebrand factor (VWF) and collagen. VWF binds to the platelet glycoprotein (GP) Ib-IX-V complex tethering platelets at the site of vessel wall injury. Collagen interacts with integrin α_2_β_1_ (also named GPIa/IIa) for adhesion and GPVI to initiate platelet activation. Soluble agonists, such as thromboxane A_2_ and adenosine diphosphate (ADP) subsequently amplify activation. Endpoints following platelet activation are characterized by: (1) shape change, (2) secretion of soluble agonists and granule content enhancing the activation process, (3) change of GPIIb/IIIa conformation to bind fibrinogen, which sustains platelet aggregation, and/or (4) externalization of negatively charged amino-phospholipids contributing to platelet procoagulant activity (Figure 1) [5,6,7]. Because of the three-dimensional configuration of the growing thrombus, platelets are differently exposed to agonists resulting in heterogeneous activation profiles [8]. Common examples of the pathophysiology are described below for each activation endpoint.

### 2.1. Adhesion

Under normal physiological conditions, the endothelium does not provide an adhesive surface for platelets. However, in the presence of vascular damage, the sub-endothelial matrix and/or layer(s) become exposed, revealing collagen and tissue factor, which are powerful haemostatic activators. The main function of platelet receptor GPIb-IX-V is to mediate the initial adhesion of circulating platelets to VWF adhered on the exposed collagen [9]. Four subunits compose the GPIb-IX-V complex: GPIbα, GPIbβ, GPIX, and GPV (encoded by four different genes *GPIBA*, *GPIBB*, *GP9*, and *GP5*) [10,11]. The N-terminal domain of GPIbα subunit has a binding site for VWF, which acts as a bridge between platelets and the fibrils of collagen in the sub-endothelial matrix and/or layer(s). This interaction is particularly important in the presence of high shear stress, in order to: i) slow down platelets in the blood stream, ii) recruit them to the site of the injury and iii) initiate the signalling cascades that will lead to platelet activation [12]. In addition to VWF, the same N-terminal domain of GPIα offers a binding site for multiple ligands, which are critical for normal or pathological haemostasis. For instance, the GPIb-IX-V complex binds to P-selectin [13] (which is present on activated platelets and endothelial cells) and to leukocyte integrin aMB2 [14], thus regulating both the recruitment of leukocytes at the site of vascular injury [15] and the complex interactions between platelets and leukocytes in thrombosis and response to inflammation [16]. In addition, the GPIb-IX-V receptor has procoagulant functions, since it mediates platelet dependent coagulation through the binding of α-thrombin, coagulation factors XI (FXI) and XII (FXII), and high molecular weight kininogen [17]. Finally, The GPIb-IX-V complex is anchored to the actin filaments of the platelets’ cytoskeleton through the binding of GPIbα cytoplasmic tail to filamin A [18]. This interaction is important for maintaining platelet shape and stability [19,20]. Defects and/or dysfunctions of this multitasker receptor have major consequences in platelet functions.

#### 2.1.1. Bernard-Soulier Syndrome

Bernard Soulier Syndrome (BSS) is an inherited bleeding disorders characterized by bleeding tendency, macro-thrombocytopenia, and defective ristocetin-induced platelet agglutination [21,22,23].

Clinical features of patients with BSS are non-specific and characterized by epistaxis, mucocutaneous and post trauma bleedings, and severe menorrhagia in females [17,24].

In most patients, BSS has an autosomal recessive pattern of inheritance, but rare forms with autosomal dominant pattern are also known [25,26]. A large number of mutations (missense, nonsense or deletions) in genes *GPIBA*, *GPIBB*, and *GP9* (but not in *GP5* [27]) have been mapped and found to be causative of BSS [17,24]. In fact, these genes (*GPIBA*, *GPIBB*, and *GP9*) are required to express efficiently the functional GPIb-IX-V complex at the platelets’ surface. In BSS platelets, the GPIb-IX-V complex is either low, absent or dysfunctional (i.e., unable to bind VWF). Thus, in BSS platelets, the normal interaction of GPIbα with VWF is abolished and platelets’ adhesion to the sub-endothelium is impaired [28]. In addition, BSS platelets show other characteristics, such as an increased membrane deformability, a poor response to low doses of thrombin, and a decreased ability to support thrombin generation [29,30,31]. All these features can be related to the absence/dysfunction of GPIb-IX-V complex [17].

The clinical suspicion of BSS has to be confirmed by different laboratory investigations. A variable degree of thrombocytopenia (platelet count range: <30 × 10^9^/L to normal [22,24]) might be observed, with a blood smear revealing abnormally large or irregularly shaped platelets (even in patients with normal platelet count) [32,33]. The closure time measured by the platelet function analyser (PFA-100/200) is increased and the bleeding time prolonged [34,35]. However, the sensitivity of PFA-100/200 assay depends on the severity of the defect [15,36], which implies further investigations (aggregometry and/or flow cytometry) to establish an accurate diagnosis. The VWF-dependent agglutination measured in the presence of ristocetin by light transmission aggregometry (LTA) is defective in homozygous BSS platelets (but normal in heterozygous form) [33]. Of note, this defect will not be rescued by the addition of normal plasma, which distinguishes BSS from von Willebrand disease (VWD) [35,37]. In vitro aggregation of BSS platelets in response to epinephrine, ADP, collagen, and arachidonic acid is normal, but a slow response is observed with low doses of thrombin [33]. The expression of GPIb-IX-V complex at the platelet surface can be assessed by flow cytometry. The specific antibody anti-CD42b directed against GPIbα is reduced or absent in BSS platelets, while the expression of CD41 (GPIIb) and CD61 (GPIIIa)–the two components of the fibrinogen receptor (also named integrin α_IIb_β_3_)–is normal [32,38]. Finally, in BSS platelets, the expression of GPIb-IX-V could apparently be normal because of the enlarged surface of the platelets, but the ratio between GPI-IX-V and GPIIb-IIIa will always be decreased compared to normal platelets [33].

#### 2.1.2. Platelet Type von Willebrand’s Disease

Platelet type pseudo-von Willebrand’s disease (PT-VWD) is a rare autosomal dominant disorder with a mild to moderate bleeding phenotype, intermittent thrombocytopenia, and enlarged platelets. 

PT-VWD is characterized by mutations in *GP1BA* [39], which enhance the affinity of the surface glycoprotein GPIbα for the VWF multimers. As a result, spontaneous binding of high molecular weight VWF to platelets occurs in vivo, leading to platelet clumping and increasing platelet clearance [40]. This causes thrombocytopenia and removal from plasma of the largest VWF multimers (which have the greatest haemostatic capacity), leading to an increased bleeding risk.

At laboratory work-up, patients with PT-VWD often have a prolonged bleeding time and platelet clumping can be observed on blood smears. The response of PT-VWD platelets to low doses of ristocetin is enhanced and the VWF multimers analysis (which assesses concentration and distribution of VWF multimers in plasma) shows loss/reduction of the largest VWF forms. PT-VWD phenotype is very similar to type 2B VWD. However, in type 2B VWD, the defect lies in the VWF molecules, which have an increased affinity for platelets. Differential diagnosis is fundamental for the correct therapy of PT-VWF or VWD 2B patients. The two conditions can be distinguished by (i) ristocetin induced platelets agglutination (RIPA) mixing experiments, (ii) cryoprecipitate challenge, and iii) flow cytometry. In the RIPA assay, washed or gel-filtered platelets from the patient are mixed with normal plasma and vice versa (i.e., normal platelets are mixed with patient plasma) in presence of low dose ristocetin. Washed/gel-filtered platelets from PT-VWD patients (but not VWD 2B platelets) will agglutinate in normal plasma (because of the abnormal GPIbα avidity for VWF characteristic of PT-VWD) and washed/gel-filtered normal platelets will aggregate in the presence of VWD 2B plasma (containing the hyper-adhesive VWF) [41,42]. Of note, in the negative control (washed/gel-filtered platelets + plasma from a healthy donor) there is no aggregation at low doses of ristocetin. The cryoprecipitate challenge [43] consists in the addition of high concentrate normal VWF to platelets, which causes PT-VWF spontaneous aggregation, but not for VWD 2B platelets; however, false positive results have been observed among VWD 2B patients [44] and this test is not included in the diagnostic algorithm proposed for PT-VWD diagnostic work-up [45]. A flow cytometry method able to highlight the increased affinity of VWF for GPIbα and to discriminate between PT-VWD and VWD 2B through mixing tests has also been proposed [46]. Finally, the identification of mutations in the *GPIBA* gene will confirm the diagnosis of PT-VWD [45].

### 2.2. Secretion

The secretion of bioactive molecules is one of the characteristics of platelet activation. Once a platelet agonist has engaged its corresponding platelet surface receptor, a signal transduction takes place, leading to a short-time increase of intracellular calcium, which promotes platelet shape change, fusion of platelet granules with the plasma membrane and consequent release of platelet contents [47]. Platelets contain three major types of granules, which are in order of abundance, α-granules (50–80/platelet), dense-granules (3–8/platelet), and lysosomes (1–3/platelet) [48]. α- and dense-granules seem to derive, like lysosomes, from multivesicular precursors [49,50,51]. The content of α-granules consists of a large variety of proteins, such as adhesive molecules (e.g., fibrinogen, VWF, fibronectin, P-selectin), coagulation factors (e.g., FV, FIX, FXIII), anticoagulants (e.g., antithrombin), fibrinolytic proteins (e.g., plasminogen), and growth factors, immune mediators, and integral membrane proteins (e.g., α_IIb_β_3_, P-selectin) [52,53]. Thus, α-granule proteins can be involved in a large spectrum of physiological functions, such as normal and pathological haemostasis, inflammation, wound healing, antimicrobial response, and cancer metastasis [54,55]. Dense-granules contain small non-protein molecules, such as nucleotides (ADP/ATP), serotonin, histamine, calcium ions (which give the dense appearance on electron microscopy), inorganic polyphosphates, membrane proteins [such as granulophysin (CD63), lysosomal-associated membrane protein 2 (LAMP-2)], [55]); plasma membrane adhesive receptors GPIb and α_IIb_β_3_ have also been identified on dense-granules by immune-histochemical studies [56]. The main function of dense-granules content is to amplify platelet activation and to sustain platelet aggregation [57]. Lysosomes store digestive enzyme involved in the degradation of proteins, carbohydrates and lipids. Their role in haemostasis and thrombosis is still unknown [55].

Platelet storage pool deficiencies (SPD) refer to a group of inherited heterogeneous disorders in which the number and/or the content of α-granules, dense-granules, or both are reduced and cannot be adequately released during platelet activation [58,59]; as a consequence, a defect mostly in ADP release from activated platelet and in secretion-dependent aggregation is observed [60]. According to the type of granule pool deficiency, the clinical syndrome is called α-SPD, δ-SPD or αδ-SPD [58,61] These anomalies are to be distinguished from secretion defects, in which granules are normally present, but abnormally secreted due to defective signal transduction or granule trafficking defects [62].

#### 2.2.1. α-Storage Pool Disease or Gray Platelet Syndrome

The Gray platelet syndrome (GPS) is a very rare disease characterized by a quantitative and qualitative deficiency of α-granules [63,64]. Patients with GPS have a mild to moderate bleeding diathesis, mild but progressive thrombocytopenia, and presence of larger and vacuolated platelets [65]. The associated phenotype is the presence of bone marrow fibrosis (due to the release of megakaryocytes in the bone marrow environment) and of splenomegaly (due to extramedullary haematopoiesis) [65,66,67]. Other features of GPS have been linked to immune dysregulation and autoimmune defects [68].

GPS megakaryocytes show a defect in α-granule production and are unable to correctly pack and store endogenous and exogenous proteins into α-granule precursors [69]. The lack of soluble proteins within α-granules, whose content is fundamental for normal haemostasis, leads to a small and unstable platelet plug [70]. The classical GPS is inherited with an autosomal-recessive pattern and it is associated with mutations or splicing alterations in *NBEAL2* gene, involved in granule trafficking [71] and retention of cargo proteins in maturing α-granules [72]. Other GPS forms with autosomal-dominant or X-linked inheritance have been reported (reviewed in [73]).

The absence of α-granules in the cytoplasm of affected platelets results in a characteristic pale or gray appearance, opposite to the purple staining of granules in normal platelets on Wright-Giemsa stained blood smears. Platelet aggregation analysis by LTA is variable: in most of the GPS patients, the responses to ADP, epinephrine, and acid arachidonic are normal, while responses to thrombin and collagen are decreased [65]. Of note, content and surface expression of P-selectin are variable and their assessment is inadequate for diagnostic purposes [65,74,75,76,77]. The diagnosis is confirmed by the lack of α-granules observed by electron microscopy and by the absence of α-granule proteins [78,79].

#### 2.2.2. δ-Storage Pool Disease

δ-Storage pool disease (δ-SPD) is a congenital abnormality characterized by a deficiency of dense-granules in megakaryocytes and platelets [79]. δ-SPD can be associated with disorders of others lysosome related organelles leading to syndromic forms, known as Hermansky-Pudlack, Chediak-Higashi, and Grisicelli syndromes, in which albinism and immune deficiency are associated with platelet function defects [80]. Patients with non-syndromic δ-SPD have a mild to moderate bleeding diathesis, mainly mucocutaneous; however life-threatening bleedings can occur after surgery or trauma [81]. Clinical presentation of δ-SPD is highly variable and so far there are no validated recommendations concerning the decisional algorithm to reach an accurate diagnosis [81], nor for δ-SPD management [82].

Patients with δ-SPD usually have normal platelet counts with a prolonged bleeding time [83]. The lack of dense-granules (and thus of ADP/ATP and serotonin) will be reflected by an impaired aggregation response to different agonists in vitro. Typically, LTA curves performed with citrated platelet rich plasma (PRP) are characterized by the absence of a second wave in ADP induced platelet aggregation and a diminished response to collagen induced aggregation (at low concentrations) [79,81]. However, a study reported that δ-SPD patients (23% of the cohort studied) had normal aggregation response [84]. Thus, further specialized tests are sometimes needed to confirm the diagnosis. In particular, whole mount transmission electron microscopy can be used to highlight the absence/reduction of dense-granules [85], while flow cytometry, by the mepacrine test uptake, is useful to evaluate the dense-granule content and secretion capacity of platelets. The mepacrine test is based on the fact that mepacrine binds to adenine nucleotides and accumulates rapidly in dense-granules. The mepacrine taken up by dense-granules is then released after platelet stimulation and fluorescence can be quantified before and after platelet activation [86,87]. δ-SPD platelets will have reduced dense-granules and low uptake and release of mepacrine [79,88,89]. Platelet content of adenine nucleotides and serotonin can be evaluated by chemiluminescence aggregometry and radio-labelled or chemical methods, respectively (reviewed in [59,81]). δ-SPD platelets will be characterized by reduced adenine nucleotides and serotonin content, with an elevated ATP/ADP ratio [90].

### 2.3. Aggregation

Platelet aggregation is mediated by the GPIIb/IIIa (integrin α_IIb_β_3_), a major receptor of the platelet surface, whose activated form binds to fibrinogen. Surface expression of GPIIb/IIIa increases after platelet activation. Upon agonist induced platelet activation, a signalling process (“inside-out” signalling) leads to conformational changes of the GPIIb/IIIa receptor, which increases its affinity for fibrinogen. The binding of fibrinogen with platelet GPIIb/IIIa receptors allows platelet aggregation (leading to the primary platelet plug), providing primary haemostasis. Binding of fibrinogen to the GPIIb/IIIa receptor initiates further intracellular signalling (“outside-in”) which induces additional granule secretion, platelet spreading, and contraction of the fibrin mesh. This signalling pathway culminates in a stable and irreversible aggregation of platelets [47,91].

#### Glanzmann Thrombasthenia

Glanzmann thrombasthenia (GT) is a rare autosomal inherited bleeding disorder, characterized by a quantitative or qualitative defect in integrin α_IIb_β_3_, also known as glycoprotein GPIIb/IIIa, which is essential for platelet aggregation and normal haemostasis.

GT is caused by mutations in the genes *ITGA2B* and *ITGB3*, which encode for subunits α_IIb_ (GPIIb, CD41) and β_3_ (GPIIIa, CD61), respectively, of integrin α_IIb_β_3_. Mutations in these genes compromise the normal function of the GPIIb/IIIa receptor, impairing platelet aggregation and interaction with its adhesive ligands and thus leading to inefficient clot formation/consolidation and to GT bleeding phenotype.

Bleeding tendency in patients with GT is highly variable and poorly correlated with the underlying genetic mutations or α_IIb_β_3_ expression level [92]. It ranges from a mild to severe haemorrhagic condition [93,94]. Typical bleeding manifestations are purpura, gum bleeding and menorrhagia, while gastrointestinal or central nervous system bleeding are less frequently reported [95]; bleeding after trauma or surgery might be severe [93,96,97]. Most patients are diagnosed in childhood, but heterozygous patients can reach adulthood being asymptomatic [93]; in general, the bleeding tendency in GT decreases with age [98]. 

GT is divided in three subtypes [93,99] according to the GPIIb/IIIa expression (determined by flow cytometry [100]) on the platelet membrane:-ߓType I, the most severe form of GT: the expression of GPIIb/IIIa is absent (<5% of normal); platelet fibrinogen and clot retraction are also absent;-ߓType II, a moderate form of the disease: surface GPIIb/IIIa is reduced with a level of expression varying between 10–20% of normal; reduced fibrinogen content and clot retraction;-ߓType III, a variant form: the expression of GPIIb/IIIa is near normal or normal (between 50–100%), but the receptor is dysfunctional; variable platelet fibrinogen content and clot retraction.

GT platelets adhere normally to the sub-endothelium, but spreading is abnormal [101,102,103]. GT platelets have decreased or absent aggregation to physiological agonists, but agglutination in response to ristocetin is normal (because it is mediated by GPIb-IX-V via VWF). Since a functional GPIIb/IIIa is required for efficient dense-granules release, in GT platelets an abnormal release might also be observed [104,105]. Laboratory findings include a normal platelet count, size and granularity, but a prolonged bleeding time [35,98]. PFA-100/200 assay shows a very prolonged closure time (>300 s), which is compatible with GT, but not specific [36,98]. LTA is considered the gold standard method for the clinical diagnosis of GT [98]. GT PRP is analysed before and after the addition of different agonists, such as arachidonic acid, ADP, collagen, and epinephrine. The absence or marked reduced aggregation in response to low or high concentrations of multiple agonists, along with a maintained response to ristocetin, indicates a defect in GPIIb/IIIa and is highly indicative of GT [36,98]. Due to variability of platelet aggregation results, it is recommended that the analysis be confirmed with a second sample [98,106,107] and to use a second round of testing with a larger spectrum of agonists [106,107]. Flow cytometry can be used to assess the quantitative deficiency of GPIIb/IIIa (GT type I and II) in the membrane of resting platelets through the use of fluorescent probes recognizing α_IIb_ (CD41) and/or β_3_ (CD61) subunits. The GT variant form, (GPIIb/IIIa expressed but not functional) can be investigated by flow cytometry using the monoclonal antibody PAC-1, which recognizes the activated form of the GPIIb/IIIa receptor after platelets stimulation. GT activated platelets will not bind with the PAC-1 monoclonal antibody, due to the dysfunctional GPIIb/IIIa receptor [107,108,109]. Finally, the identification of the specific mutation variants in *ITGA2B* and *ITGB3* genes is the key to a complete diagnosis of GT [98,108].

### 2.4. Procoagulant Activity

Following strong activation, platelets expose negatively charged phospholipids on their outer membrane. This is essential in order to achieve an efficient haemostatic response by generating high amounts of thrombin and subsequent clot stabilization by fibrin. This peculiar platelet feature and its clinical role and relevance will be extensively described in the second part of this review.

## 3. Expression of Negatively Charged Phospholipids and Their Role in Coagulation

At resting state, the phospholipids of the cell membrane are asymmetrically distributed, thanks to flippase/floppase activity [110]. Neutral phospholipids (e.g., phosphatidylcholine, sphingomyelin, and sugar-linked sphingolipids) are located on the external leaflet of the membrane, while negatively charged phospholipids (phosphatidylserine (PS) and phosphatidylethanolamine) are within the inner surface of the membrane.

Under specific circumstances, such as apoptosis or strong platelet activation, this distribution is altered. During platelet activation, scramblases (such as TMEM16F, also known as anoctamin 6) shuffles the phospholipids between the two layers, resulting in the expression of PS on the external leaflet [110]. Despite similar endpoints, apoptotic-induced and agonist-induced PS exposure are two distinct pathways, both resulting in PS exposure (reviewed in [111]).

Apoptosis is a slow process (taking hours) that results with platelet aging and is mediated through the activation of caspases, pro-apoptotic Bak/Bax-mediated mitochondrial collapse, and PS exposure (mostly TMEM16F-independent) [112]. This slow process leads to platelet clearance.

Strong platelet activation induces a rapid (one–two minutes) necrotic-like phenotype via elevated and sustained cytosolic calcium concentration, mitochondrial depolarization, calpain activation, and TMEM16F-dependent PS exposure [113,114]. Plasma membranes form a small “cap” area enriched in exposed PS [115]. Such micro-domains concentrate blood coagulation factors and accelerate enzymatic reactions.

Indeed, in synchrony with platelet activation and aggregation, fibrin deposition is an important process for the stabilization of the haemostatic clot [116]. This is achieved by thrombin cleaving fibrinogen into fibrin as a consequence of a series of sequential reactions engaging activated coagulation factors, in which calcium and negatively charged phospholipids are critical mediators [117].

Some coagulation factors (factors II, VII, IX, X) experience vitamin-K dependent posttranslational ɣ-carboxylation of C-terminal glutamic acid residues [118,119]. These highly negative domains confer to factors high-affinity binding for calcium, which facilitates their interaction with negatively charged phospholipids. In fact, activated coagulation factors interact poorly with each other in solution. Calcium binding is instrumental for supporting binding of coagulation factors to a membrane of negatively-charge phospholipids, such as the surface of procoagulant platelets [120,121].

In addition to rapid phospholipid membrane remodelling and PS externalization, platelet procoagulant response is accompanied by the release of microparticles from the membrane surface of activated platelets [122,123]. The mechanisms underlying the formation of platelet derived microparticles (PMPs) involve the increase of cytoplasmic calcium affecting the activity of intracellular enzymes, the phospholipid transient mass imbalance between the two leaflets of the membrane, and the proteolytic action of calpain on the cytoskeleton [124]. PMPs shed from activated platelets provide a source of supplementary negatively charged surface on which blood coagulation factors can assemble, thereby enhancing the procoagulant response [122]. Dale et al. [125] showed that the number of PMPs produced by procoagulant platelet was higher than the number of PMPs produced by aggregating platelets but 5.4 times lower than PMPs originating from A23187 calcium ionophore activated platelets. Sinauridze et al. [126] studied the procoagulant properties of A23187-calcium ionophore activated platelets and PMPs. The authors showed that the surface of PMPs originated after A23187 activation is 50- to 100-fold more procoagulant than the surface of activated procoagulant platelets. This stronger procoagulant activity was related to a higher density of procoagulant phospholipids on PMPs’ membrane. From a physiological point of view, the observation that procoagulant collagen-and-thrombin (COAT) platelets produce less PMPs than ionophore does [125,127], might indicate that COAT platelet dependent thrombin generation (TG) should be contained at the site of vascular injury to avoid an unnecessary and dangerous systemic spread.

Taken together, the phospholipid surfaces enhance the enzymatic function of coagulation factors [128]. Membrane binding and surface diffusion facilitate and accelerate the encounter of coagulation partners (e.g., the assembly of tenase and prothrombinase complexes) [128]. This facilitates the rate of activation of prothrombin by several orders of magnitude. Therefore, the platelet contribution has a considerable impact on the procoagulant response, by localizing and enhancing thrombin generation directly at the site of vascular wall damage.

## 4. Procoagulant Platelets

Following strong activation, a fraction of platelets expresses PS on their surface and become highly efficient in sustaining thrombin generation.

Since the first descriptions in the late 1990s, several synonyms have been used (extensively described in recent reviews [129,130]) such as collagen-and-thrombin (COAT)-activated platelets [87,127,131], COATed platelets [132,133], ballooned and procoagulant platelets (BAPS) [134], sustained calcium-induced platelet morphology (SCIP) platelets [135], super-activated platelets [136], super platelets [137] and even zombie platelets [138,139]. Despite this diverse classification, they all share the very same characteristics of necrotic-like mechanisms [111,140], leading to procoagulant activity through expression of PS [130].

In particular, after strong activation, all platelets display an initial cytosolic calcium increase and GPIIb/IIIa activation [131]. However, after a certain delay (1–2 min), while aggregating, platelets decrease their calcium level, and procoagulant platelets reach higher cytosolic calcium concentration [131,141,142]. In addition to calcium mobilization from intracellular stores and store-operated calcium entry, calcium influx mediated by sodium-calcium exchanger (NCX) reverse mode is critical for achieving the high calcium level required to trigger the formation of the mitochondrial permeability transition pore (mPTP), leading to cyclophilin D-dependent mitochondrial depolarization [141,142,143]. This results in very high and sustained cytoplasmic calcium, gradual inactivation of GPIIb/IIIa receptors [131,144], activation of TMEM16F [113], and PS externalization [114,134], which eventually induces the procoagulant activity of platelets together with microparticle generation [47,127,134,145].

In addition to the procoagulant activity mediated through PS exposure, procoagulant platelets gain pro-haemostatic function by retaining α-granule proteins on their membranes, such as coagulation factor V/Va, fibrinogen, VWF, thrombospondin, fibronectin, and α2-antiplasmin in a serotonin- and transglutaminase-dependent mechanism [146].

### 4.1. Clinical Features of Procoagulant Platelets

The potential generation of procoagulant platelets is on average ca. 30% in healthy donors, with a wide range from 15–57% described in the literature [87,132,147,148]. In our laboratory, we have a mean of 38.9% (SD 8.3; range 21.9–59.1%, *n* = 73) ([149] and Adler et al., manuscript in preparation). However, despite a wide inter-person variability, the individual values are stable over time [132].

Clinical interest in procoagulant platelet potential has largely increased during the last two decades. Especially, stratification of this wide range could associate extreme values to clinically relevant medical situations, such as in haemostatic imbalances (bleeding or thrombotic events) or even in non-haemostatic circumstances.

#### 4.1.1. Low Level of Procoagulant Platelets Is Associated with Impaired Platelet Function and Bleeding Diathesis

The Scott syndrome was the first clinically relevant bleeding disorder associated with impaired platelet procoagulant activity [150]. In this very rare congenital bleeding disorder, patients have impaired phospholipid scrambling and do not expose PS at the membrane surface even after treatment with calcium ionophores [151,152]. Besides this complete absence of PS exposure, a reduced ability to generate procoagulant platelets has been shown to increase bleeding risk. Of note, low levels of procoagulant platelets (<20%) were detectable in about 15% of patients with a clinically relevant bleeding diathesis and an unrevealing standard work-up, including LTA and secretion assays ([87,153] and Adler et al., manuscript in preparation).

Moreover, patients with spontaneous intracerebral haemorrhage have a significantly lower percentage of procoagulant platelets compared to controls (24.8 ± 9.7% vs. 32.9 ± 12.6%) [154]. In a similar cohort of patients, those who generated the lowest levels of procoagulant platelets encountered more severe haemorrhages with increased bleed volumes [155] and, in another study, patients with procoagulant platelet levels lower than 27% had a poor outcome and increased mortality at 30 days [156]. Similarly, patients with subarachnoid haemorrhage that generate procoagulant platelets in the lowest range of the cohort (<36.7%) faced an increased mortality rate after one month [157]. However, these patients had on average a higher level of procoagulant platelets compared to controls (41.8 ± 11.4% vs. 30.7 ± 12.2%). As discussed by the authors, this antithetical observation could be related to the presence of a chronic inflammation in this pathology (but whether inflammatory state amplifies the procoagulant activity or the other way around is difficult to clarify; see below).

Interestingly, even in some cerebral thrombotic pathologies, patients who generated procoagulant platelets in the lowest range of the cohort presented increased bleeding phenotypes, with more microbleeds [158] or early secondary bleeding into the ischemic brain area compared to the other patients from the same cohort [159].

Discordant observations were reported regarding platelet procoagulant potential in two cohorts of haemophilia A patients. Both studies reported a reduced potency in generating procoagulant platelets compared to controls [160,161]. However, while Saxena et al. [160] observed a significant difference of procoagulant platelet levels in relationship to the phenotype severity, this was not replicated by Lastrapes et al. [161]. 

A single study also reported an impaired ability to generate procoagulant platelet in patients with essential thrombocythemia compared to controls and this was rescued by hydroxyurea treatment [162]. 

#### 4.1.2. High Level of Procoagulant Platelets Worsens Thrombotic Events

In contrast to the findings in bleeding phenotypes, it was demonstrated that patients with prothrombotic states had a higher potential to generate procoagulant platelets.

Mean levels of procoagulant platelets were elevated in patients with cortical strokes [163] or transient ischemic attack (TIA) [164]. Moreover, the stratification of procoagulant platelet levels increased their prognostic value. Higher levels of procoagulant platelets at the time of the cortical strokes (>34%) or TIA (>51%) were associated in both conditions with an increased incidence of stroke recurrences [165,166]. In patients with symptomatic large-artery disease, procoagulant platelet levels in the highest range of the cohort (≥50%) were associated with a higher risk for early ischemic events [167]. Similarly, for patients with asymptomatic carotid stenosis, higher levels of procoagulant platelets (≥45%) predicted a risk for stroke or TIA [168].

Contrary to the other brain ischemic situations, data showed lower mean levels of procoagulant platelets following lacunar stroke compared to non-lacunar or control levels [163]. Nevertheless, patients with higher procoagulant platelet levels (≥42.6%) experienced more recurrent ischemic events following lacunar stroke [169].

In addition to brain infarction, a high level of procoagulant platelets was also observed in coronary artery disease and heart failures [170,171,172]. 

Monitoring of procoagulant platelet potential, following an acute event, may also predict severe outcomes. A significant rise of procoagulant platelet generation after aneurysmal subarachnoid haemorrhage predicted delayed cerebral ischemia and worsening of cognitive and physical outcomes [173,174].

Higher mean levels of procoagulant platelets were also measured in cigarette smokers compared to non-smokers [147,169,175]. This is of particular interest as smoking is widely associated with an increased risk factor for cardiovascular diseases. Interestingly, smoking cessation was observed to lower the procoagulant platelet levels for individuals who quit smoking after a stroke in comparison to those who continued smoking [176]. 

#### 4.1.3. Procoagulant Platelets in Non-Haemostatic Pathologies

Massive haemorrhage in trauma is a leading cause of morbidity and mortality. Interestingly, it was recently reported that these patients experienced an increase in circulating procoagulant (balloon-like) platelets, which is in line with an increased ability to generate thrombin and a reduction of platelet aggregation [177]. This work highlights that trauma contributes to the increase of the procoagulant phenotype by the release of histone H4 from injured tissues, and, very interestingly, the authors could identify a platelet procoagulant phenotype that is already present in vivo, in contrast to other studies where the procoagulant ability of platelets is usually appreciated with ex vivo stimulations.

Interestingly, procoagulant platelets are also able to retain full-length amyloid precursor protein on their surface [178]. Further studies related levels of procoagulant platelets with Alzheimer disease severity and progression. Higher levels of procoagulant platelets were measured in early stages of the disease [179], among patients with the most severe decline [180], and among amnesic subtypes of patients with mild cognitive impairment with a progression to Alzheimer disease [181,182].

High levels of procoagulant platelets were observed in patients with end-stage renal failure [183]. Authors associated this with an increased inflammation state, but the role of procoagulant platelets as marker or trigger of thrombosis in this situation needs further investigations. Moreover, the direct influence of inflammation on procoagulant platelets (or vice versa) is not fully understood and dissecting this clearly remains challenging. Of note, inflammation is able to directly activate the haemostatic system [184] and some authors reported a relationship between high levels of procoagulant platelets and inflammation or immune system activation [132,147,183]. However, necrotic-like phenotypes, such as in procoagulant platelets, are also known to activate inflammation and immune cells [111,185].

In transfusion medicine, a low level of procoagulant platelets was observed in platelet concentrates from apheresis (16%) [186], buffy-coat (8%) [187], or cryopreserved platelet concentrates (17%) [188].

### 4.2. Pharmacological Modulation of Procoagulant Platelets

Platelets play a very important role in arterial thrombosis. Various antiplatelet therapies have been developed to prevent thrombotic events. However, these drugs aim at inhibiting platelet aggregation and, thus far, poor attention has been paid to platelet procoagulant activity.

On the other hand, different clinically relevant pharmacologic molecules have already been shown to modulate generation of procoagulant platelets. 

#### 4.2.1. Antiplatelet Drugs

Aspirin (acetyl-salicylic acid) is universally used as a standard for secondary prevention of recurrent arterial ischemic events. It irreversibly acetylates the active site of cyclooxygenase-1 (COX-1), required for the production of the soluble platelet agonist thromboxane A2. Chronic use of aspirin reduces the levels of procoagulant platelets in individuals [140,147,176]. However, intermittent or short-term uses do not relevantly impact potency in generating procoagulant platelets. While long-term use of aspirin appears to have an effect on megakaryocyte physiology inducing impaired platelet function, the direct interference with thromboxane A2 signalling does not seem to have a direct impact on the generation of procoagulant platelets [189].

ADP is able to augment the procoagulant potential induced by combined platelet activation with strong agonists, such as collagen and thrombin [187,189,190]. Accordingly, inhibition of P2Y12 (but not P2Y1) with clopidogrel [176,190] and cangrelor [191] reduces the generation of procoagulant platelets [189]. A similar effect was observed in vitro with the active metabolite of prasugrel [192].

Some of the data is sparse on the in vitro use of antagonists of the GPIIb/IIIa and the effect on procoagulant platelets. One study demonstrated that pre-treatment with either eptifibatide, tirofiban, or abciximab augmented the potential to generate procoagulant platelets [193]. This could explain the failure of long-term use of oral GPIIb/IIIa-antagonists observed in the early 2000s [194]. However, the procoagulant potentiation obtained with GPIIb/IIIa-antagonists was not corroborated by others [149,195,196,197]. These discordant data were all obtained with in vitro pre-treatment. Directly investigating the ability to generate procoagulant platelets in patients under treatment with GPIIb/IIIa-antagonists would help to clarify these discrepancies.

#### 4.2.2. Off-Target Procoagulant Platelet Modulation

Desmopressin (1-deamino-8-D-arginine vasopressin (DDAVP)), a synthetic analogue of vasopressin initially used to treat diabetes insipidus and enuresis nocturna, improves the haemostatic status of patients by raising plasma levels of VWF and coagulation factor VIII [198]. In addition, it has also been demonstrated in vitro that DDAVP is a weak inducer of procoagulant response of platelets [199] as well as arginine vasopressin [200]. This was corroborated with in vivo treatment of patients with mild platelet disorders [201]. In this study, DDAVP was able to increase generation of procoagulant platelets by enhancing calcium and sodium mobilization. A similar observation was made in cardiac surgery patients receiving DDAVP because of postoperative excessive bleeding [202].

Auranofin, a thioredoxin reductase inhibitor used to treat rheumatoid arthritis was reported to induce calcium overload and increased oxidative stress in platelets, which would contribute to a necrotic PS exposure [203]. 

Patients using selective serotonin reuptake inhibitors (SSRI) had significantly lower procoagulant platelet levels compared to individuals not taking SSRI [147]. Furthermore, citalopram, a SSRI, was demonstrated to impair GPVI-mediated platelet function [204]. This is supported by the importance of serotonin for the formation of procoagulant platelets [146,205] and the mild bleeding diathesis reported in patients under SSRI treatment [206].

Inhibition of the procoagulant response of platelets was also observed with tyrosine kinase inhibitors used in oncology [207,208,209,210]. These pharmaceuticals reduce formation of procoagulant platelets by inhibiting tyrosine signalling downstream of GPVI activation.

### 4.3. Laboratory Work-Up for Investigating Procoagulant Platelets

Procoagulant platelets can be easily detected and characterized in vitro with fluorescence labelling and therefore by using microscopy or flow cytometry. Flow cytometry assays allow quantification of the ability to generate procoagulant platelets (see above, Section 4.1 and Section 4.2) and to analyse phenotypically different platelet subpopulations. Moreover, flow cytometry is an accessible, easy, and rapid diagnostic tool for haematological diagnostic laboratories. Procoagulant activity can be appreciated as well with other assays, such as ex vivo platelet-dependent thrombin generation and flow chambers. However, these latter techniques are for now experimental methods and their diagnostic utility still needs more investigations. Finally, in vivo assays with animal models are also of high interest to study the thrombus distribution of procoagulant platelets and to understand better physiological and pathophysiological thrombus formation.

#### 4.3.1. Quantification and Characterization of Procoagulant Platelets

Table 1 summarizes the main procoagulant activation endpoints and the markers used to detect and to discriminate the procoagulant platelet subpopulation, commonly used for flow cytometry. Surface expression of PS is the major standard activation endpoint widely recognized for procoagulant platelets. The gold standard assay to detect this event resides in the ability of the platelets to bind Annexin V [87,127] or lactadherin [211,212,213]. Another necrotic-like event that occurs in procoagulant platelets is the loss of the mitochondrial potential. This cytoplasmic event can be detected with mitochondrial probes like rhodamine derivatives, such as tetra-methyl-rhodamine methyl ester (TMRM) or tetra-methylrhodamine ethyl ester (TMRE) [131,142,214] or the carbocyanine JC-1 [140]. Rhodamine probes accumulate into intact mitochondria, but once platelets experience loss of the mitochondrial membrane potential, they escape and fluorescence decreases [215]. The JC-1 probe naturally exhibits green fluorescence. Its accumulation into intact mitochondria induces formation of probe aggregates that induce a fluorescence emission shift from green to red. Therefore, the red/green fluorescence intensity ratio is an indicator of the mitochondrial potential allowing the detection of mitochondrial depolarization by a decrease in the red/green fluorescence ratio [215].

Because procoagulant platelets lose their properties to aggregate, the PAC-1 binding assay is another interesting approach to discriminate procoagulant platelets from non-coagulant aggregating platelets [131,142,216,217].

Last but not least, procoagulant endpoint is the coating of α-granule proteins on the surface of procoagulant COAT platelets [127,146,218,219]. This approach relies on the analysis of the surface retention of α-granule proteins with specific monoclonal antibodies. This technique is not often employed by clinical diagnostic laboratories, but can be performed in research laboratories, as it requires a specialized method and technical expertise to detect it properly. 

#### 4.3.2. Assessment of the Overall Coagulation Potential and Procoagulant Activity of Platelets

An arsenal of different complementary methods, which we have briefly summarized in Table 2, are available to assess the procoagulant potential in biological samples. The procoagulant activity of PS expressed by platelets and PMPs can be directly measured in plasma by functional tests (clot or chromogenic based assays), which take advantage of the anionic phospholipid dependent acceleration exerted by PS on prothrombin activation by the FXa-FVa complex [220,221].

Thrombin generation assay (TGA) is a sophisticated technique capable of assessing the delicate balance of procoagulant and anticoagulant pathways involved in the haemostatic process, thus providing a global view of the coagulation potential of an individual. The standard reference method for measuring thrombin generation (TG) is the calibrated automated thrombogram (CAT) developed by Hemker [222]. TGA can be performed using various types of biological material: most commonly, the assay is performed in PRP or platelet poor plasma (PPP). PRP is useful to study the interaction of platelets with coagulation factors in the coagulation process. Working with PPP requires the addition of artificial phospholipids to the sample (as substitute for platelets in order to provide the negatively charged surface that sustains TG); PPP investigation focuses on the action of coagulation factors. A particular advantage of PPP is that the sample can be frozen (thus allowing storage) and thawed just before analysis. The measurement is performed in the presence of defined concentrations of tissue factor (low, normal or high), allowing the modulation of the sensitivity of the test (e.g., high concentration of tissue factor will make the test less sensitive to the intrinsic pathway). Thrombomodulin-modified TGA is a novel variant of the classical TGA, which allows the highlighting of the role of the protein C system in downregulating the coagulation process [223]. This might be of interest for investigating platelet-dependent TG because it has been demonstrated that platelet-derived activated coagulation factor Va (FVa) bound on the surface of procoagulant platelets is protected from inactivation catalysed by activated protein C [224]. Finally, interesting and innovative technologies based on a spatio-temporal model of haemostasis, have been used to measure the contribution of procoagulant platelets or PMPs to the growth of the fibrin clot [126].

A step closer to physiological coagulation is represented by ex vivo TG measurement in whole blood. However, this method is challenging due to the interference of erythrocytes on the stability of fluorescence signal and requires expert operators. An alternative method to overcome the problem of the turbidity or colour of the blood sample is based on monitoring TG by electrochemistry. Such a method was developed by Thuerlemann et al. [225] using a single-use electrochemical biosensor sensible to the electric variations produced by an amperogenic substrate cleaved by thrombin. The variation of electric signal is recorded and the raw data values used to build a TG curve.

To exclude the effect of plasmatic factors, platelets can be isolated by gel filtration [201] or washing steps [131]. The specific contribution of procoagulant platelets to TG can be assessed by modified TG assays [126,201]. Gel filtered/washed platelets, once activated with specific agonists to the procoagulant phenotype, also generate procoagulant PMPs. The latter can be directly identified and investigated by flow cytometry based on their size (FSC) and specific fluorescent dye binding to exposed PS [125]. Flow cytometry is a powerful and preferred technique for investigating PMPs [226], since it allows counting, identifying their origin, and determining PS exposure by Annexin V binding [227]. Drawbacks of PMP measurements with flow cytometry are the small and heterogeneous size (0.1 to 1 μm) of PMPs, which can be very close to the instrument background and the difficult of calibration. It is possible to overcome these limitations by using fluorochrome tagging PMPs (e.g., molecules incorporating the phospholipid bilayer) and size-calibrated fluorescent beads together with background noise reduction (through 0.1 μm liquid filtration). Nevertheless a good expertise and high resolution flow cytometers are required [227]. PMPs generated from procoagulant platelets can be further processed to obtain a pure PMP preparation by subsequent centrifugation steps and used to measure PMP-dependent TG [126].

#### 4.3.3. In-Vivo Investigations of Procoagulant Platelets

Intravital microscopy permits the study of physiological haemostasis and the appreciation the heterogeneous structure of a growing thrombus [243,244]. More and more publications are present in the literature assessing the heterogeneous platelet activation status with a particular focus on the role of procoagulant platelets [134,245,246]. Very recently, Nechipurenko et al. demonstrated that, during the in vivo formation of the thrombus, the procoagulant platelets are located at the periphery of the clot, which is driven by their mechanical extrusion as a result of the clot contraction [247]. These increasing new data provided by intravital microscopy and future experimentation with genetically-engineered mouse models, such as TMEM16F-deficient mice [246], will increase our knowledge on the in vivo role of procoagulant platelets. Obviously, this can be extended to other thrombocytopathies, where we can also obtain a real time monitoring of thrombus formation in pathophysiological conditions [248,249,250].

Nevertheless, one should be aware that such experiments still lack standardization, and inter-laboratory replicability is laborious. We should also keep in mind that even though this technique allows a step closer in studying haemostasis and thrombosis, experiments have thus far been performed with non-physiological injuries and in murine models.

## 5. Thrombocytopathy Associated to COVID-19

The current ongoing outbreak of coronavirus disease 2019 (COVID-19) is caused by a viral infection from severe acute respiratory syndrome coronavirus 2 (SARS-CoV-2) [251]. Even though SARS-CoV-2 infection initially results in excessive inflammation and mild to acute respiratory distress syndrome, patients also experience a hypercoagulable state characterized by immuno-thrombosis [252,253]. Therefore, venous and arterial thrombotic complications are an important cause of morbidity and mortality in COVID-19 patients [254].

Although the research on mechanisms implicated on platelet dysfunction in COVID-19 is still ongoing, at the time of this review there is some emerging evidence of COVID-19-associated thrombocytopathy [255,256,257,258]. In addition to a mild thrombocytopenia, which is frequent among COVID-19 patients, studies have described altered platelet function and reactivity [257,259,260,261].

Platelets seem to circulate in an activated state as demonstrated by a higher expression of specific platelet activation markers, such as P-selectin (CD62P), LAMP-3, and GPIIb/IIIa in unstimulated platelets from COVID-19 patients compared to healthy controls [259,261,262]. Platelets from SARS-CoV-2 infected patients increased basal reactive oxygen species (ROS), but basal surface expression of PS was not altered [261,263].

In addition, platelets from COVID-19 patients are hyper-responsive. Platelets had increased aggregation response to subthreshold concentrations of agonists, as well as increased adhesion and spreading [259,260,261]. This could be linked to the observed increased expression of adhesive receptors, such as VWF- and fibrinogen-receptors, respectively GPIbα/GPIX and GPIIb/IIIa [259]. Of note, COVID-19 patients had a reduced procoagulant platelet response ex vivo [263]. This was observed with a reduced mitochondrial depolarization and externalization of PS, compared to controls.

Mechanisms leading to thrombocytopathy in COVID-19 still need to be understood. However, based on the literature, platelet hyper-responsiveness may be induced by increased circulating VWF (endothelial injury) [264], hypoxia [265,266,267], and/or a hyperinflammatory environment with high cytokine levels [268,269,270], and increased oxidative stress [271].

On current observations, it seems that procoagulant platelets should not contribute to the pathophysiology of COVID-19 patients, but the hyperreactive adhesion and aggregation may be implicated.

## 6. Conclusions

Thrombocytopathies are a diagnostic challenge. The introduction of flow cytometry, as an extension to routine diagnostic work-up by LTA and secretion assays, greatly improved management of patients with a bleeding diathesis in whom previous laboratory analysis could not identify a cause [87]. Moreover, in addition to the traditional platelet aggregation assays, flow cytometry has the advantage of rapidly acquiring intrinsic properties from thousands of single platelets, of requiring small blood volumes thus enabling the analysis of samples from thrombocytopenic patients, and the exploration of more than only one endpoint of the heterogeneous profiles, as performed with traditional aggregation assays. Flow cytometry is therefore able to point out surface membrane receptor deficiencies, such as BSS (adhesion endpoint) or GT (aggregation endpoint), as well as secretion endpoint defects (dense granule content and secretion by means of mepacrine, or alpha-granules, by investigating e.g., VWF content or surface expression of P-selectin). Finally, as highlighted in this review, flow cytometry is also able to cover the important procoagulant aspect of the pleomorphic platelet activation endpoints.

Wide systematic investigation of the procoagulant activity of platelets is increasingly described in the literature. This accumulating evidence indicates that the ability to generate procoagulant platelets at and beyond the extremes of the wide normal reference range [87] is associated with clinically relevant bleeding or thrombotic disease. Specifically, the generation of procoagulant platelets at levels <20% or >50% seems to worsen bleeding or thrombotic episodes, respectively. Moreover, the individual potential to generate procoagulant platelets at the time of the clinical event (e.g., stroke) seems to be strongly related to prognosis. It remains to be investigated whether an individual baseline potential to generate high or low level procoagulant platelets would also be a risk stratification for cardiovascular diseases before their clinical manifestation.

However, most of the publications were monocentric pilot studies and/or performed with relatively small cohort sizes and/or with short follow-up timeframes. The flow cytometric investigation of procoagulant platelets still needs standardization to allow proper meta-analysis and generalization of its use. In parallel, future research and experimentation on the procoagulant status of platelets and in vivo thrombus formation models will help to better appreciate the crucial role of procoagulant platelets in haemostatic diseases. These approaches will help to dissect the role of procoagulant platelets in thrombotic and haemorrhagic events.

## Figures and Tables

**Figure 1 jcm-10-00894-f001:**
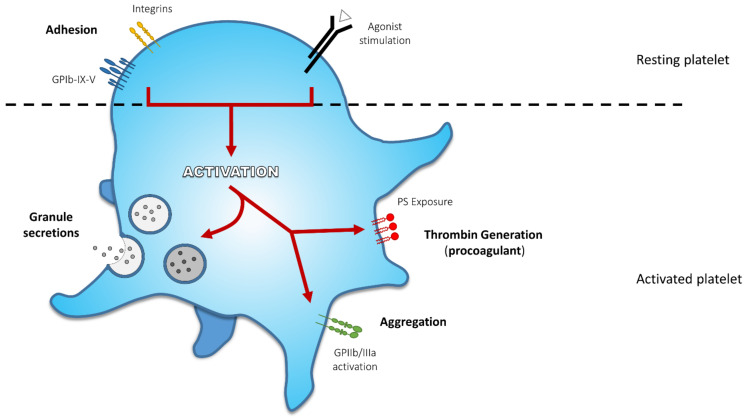
Principal Activation Endpoints During Platelet Activation. At first, platelet receptors interact with adhesive agonists exposed at the site of lesion: von Willebrand factor (VWF) binds to glycoprotein (GP) Ib-IX-V complex and collagen interacts with integrin α_2_β_1_ for adhesion and GPVI to mediate platelet activation. These first interactions initiate platelet response. Soluble agonists released by either activated platelets or injured tissue amplify platelet response and activation. These agonists induce proper receptor activation and their signalling converge to activate a core set of intracellular signalling pathways leading to various activation endpoints, such as shape change and formation of pseudopodia, secretion of α-granule and dense granule content, activation of GPIIb/IIIa sustaining platelet aggregation, and externalization of negatively charged amino-phospholipids, contributing to platelet procoagulant activity (thrombin generation).

**Table 1 jcm-10-00894-t001:** Activation endpoints of procoagulant platelets and common flow cytometry markers to detect and discriminate them.

Endpoint	Description	Common Markers	Phenotype in Procoagulant Platelets	Phenotype in Non-Procoagulant Platelets
Necrotic-like				
Phosphatidylserine	Negatively charged amino-phospholipids of platelet membrane bilayer, contribute to the procoagulant activity	Annexin V, lactadherin	Positive	Negative
Mitochondrial membrane depolarization	Mitochondrial events (depolarization) are implicated in platelet procoagulant activity process	Rhodamine (such as TMRM)	Low TMRM staining	High TMRM staining
		JC-1	Lower JC-1 fluorescence ratio (red/green)	Higher JC-1 fluorescence ratio (red/green)
Fibrinogen receptor GPIIb/IIIa (integrin α_IIb_β_3_)	Platelet membrane glycoprotein; in its activated conformation binds to fibrinogen and mediates platelet aggregation	Anti-CD41/CD61 IgM antibody recognizing the activated conformation (PAC-1)	Negative	Positive
Platelet surface coating by α-granule proteins	Proteins present in α-granule secreted upon platelet activation and retained on the platelet surface by a serotonin- and transglutaminase mechanism	Specific antibodies against α-granule proteins, such as FV/Va, fibrinogen, VWF, fibronectin, thrombospondin, and α2-antiplasmin	Positive	Negative

Legend: FV, coagulation factor V; FVa, activated coagulation factor V; TMRM, tetra-methyl-rhodamine methyl ester; VWF, von Willebrand factor.

**Table 2 jcm-10-00894-t002:** A non-exhaustive list of techniques to assess coagulation potential and procoagulant activity.

Type of Sample	Assay What Does It Measure?	Assay Name and Principle	Advantages	Disadvantages	References
WB	Coagulation potential (subsampling TG measurement)	TGA chromogenic	Presence of all blood cells and coagulation factors	Tedious subsampling at interval points; Time consuming; Only a snapshot picture of TG is available	[228]
	Coagulation potential (continuous TG measurement)	TGA Paper based WB-TG assay Fluorogenic (rhodamine 110-based thrombin substrate)	Close to physiological haemostasis; Presence of all blood cells and coagulation factors	Potential of procoagulant platelets is not specifically targeted; Calibration is difficult because of haemolysis and/or haematocrit might vary in WB sample; Interference of contact activation; Needs experienced operator	[229,230]
		TGA Novel WB-TG assay Fluorogenic (rhodamine 110-based thrombin substrate)	Close to physiological haemostasis; Presence of all blood cells and coagulation factors; Stable light transmission achieved by continuous mixing of the assay plate	Potential of procoagulant platelets is not specifically targeted;	[231]
PRP	Coagulation potential (continuous TG measurement)	TGA e.g., Thrombinoscope (Stago), Techno-thrombin (Techno-clone) Fluorogenic	Mimics in vivo condition; Consider the interaction of platelets and coagulation factors	Potential of procoagulant platelets is not specifically targeted; Standardization is difficult; Reactivity of platelets: easy to provoke unwanted activation	[232]
PPP	Coagulation potential (continuous TG measurement)	TGA e.g., Thrombinoscope (Stago), ST Genesia (Stago) Fluorogenic	Defined concentration of tissue factor and artificial phospholipids; Standardization possible in automated version; Possible to store frozen samples	Potential of procoagulant platelets is not specifically targeted; Do not consider the interaction of platelets with coagulation factors; Loss of sensitivity for the intrinsic pathway if high amount of TF is used	[222,233]
		TM-TGA ST Genesia (Stago), Fluorogenic	To study the role of protein C system by comparison of TM− and TM+ samples	TGA automated version: exact tissue factor concentration is not communicated	[223,234]
	Spatio-temporal dynamics of coagulation (real time TG and fibrin clot formation)	Thrombodynamics Video microscopy system based on measurements of light scattering images intensity	Pre-analytics is standardized; TG and fibrin formation measured at the same time; Allows to investigate separately TF-dependent and TF-independent coagulation; PRP can be added to the mix	Problematic with lipemic samples; Available only in specialized laboratory	[235,236]
Gel filtered or washed platelets	Coagulation potential (continuous TG)	Modified TGA assay fluorogenic	Targets specific procoagulant populations	Preparation is laborious; Requires experienced operator	[126,201]
	Quantifies the number of procoagulant platelets	Flow cytometry fluorescence	Targets procoagulant platelet formation and associated markers		[131]
	Measures the rate of clot growth	Experimental video microscopy Based on intensity of light scattering images	Specifically assess the contribution of activated platelets to clot growth	Requires experienced operator	[126]
PMPs	Quantifies procoagulant potential of PMPs expressing PS.	Zymuphen MP Activity assay (Hyphen BioMed) ELISA, chromogenic	Easy to perform; High speed of sample analysis	Size of the PMPs can affect binding to Annexin V, thus lower detection of PS; No information on count, size or origin	[220,221,237,238]
	Procoagulant potential of PMPs expressing PS added to phospholipid free plasma	Procoag PPL (Stago) Clotting time Number of PMPs is inversely proportional to clotting time	Can be used also on WB, PRP, PPP; Easy to perform	No information on count, size or origin	[239,240,241]
	Quantifies PMPs derived from gel filtered/washed platelets	Flow cytometry fluorescence Identification of PMPs by size (FSC) and fluorescence (e.g., bodily-label)	Target PMPs derived specifically from procoagulant platelets; Gel filtration/washing remove plasmatic components	PMPs are close to electronic noise and debris, part of the population might be below the threshold Require expertise and sensitive cytometer	[125,242]
	Coagulation potential (continuous TG)	Modified TGA Fluorogenic Isolation of PMPs by centrifugation	Specifically assess contribution of PMPs derived from procoagulant platelets to TG	Preparation is laborious	[126]
	Measures the rate of clot growth	Experimental video microscopy systemBased on intensity of light scattering images	Specifically assess the contribution of PMPs isolated from activated platelets to clot growth	Require experienced operator	[126]

Legend: ELISA, enzyme linked immunosorbent assay; FSC, forward scatter; PMPs, platelet derived microparticles; PPL procoagulant phospholipid; PPP, platelet poor plasma; PRP, platelet rich plasma; PS, phosphatidylserine; TGA, thrombin generation assay; TG, thrombin generation; UFP, ultra-centrifuged plasma; WB, whole blood.

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
