# Peer review of "Thrombocytopathies: Not Just Aggregation Defects—The Clinical Relevance of Procoagulant Platelets"

_jcm, 2021, doi:10.3390/jcm10050894_

Round 1

Reviewer 1 Report

In this review, the authors descibe several forms of inherited platelet disorders and procoagulant platelets.

In my opinion, this review does not add to our current knowledge. Their are many reviews available on inherited platelet disorders, and this one does not stand out. The review on procoagulant platelets is not new enough to merit an separate publication.

I have a feeling this is more a general introduction for a thesis.

Reviewer 2 Report

The manuscript of the review titled Thrombocytopathies: Not just aggregation defects – The clinical relevance of procoagulant platelets by Aliotta and Bertaggia Calderara et al. is an excellent example scholarly work. 

The authors offer a comprehensive insight into the physiological background, clinical relevance, and laboratory testing of thrombocytopathies affecting platelet adhesion, secretion, and aggregation. Besides, I also found the review to be a much needed, timely summary of the extensive literature on thrombocytopathies. 

I do not have any major suggestions for the authors; however, if there is no word limit, could it be possible for the authors to add a short section on thrombocytopathies in COVID19? This will be an excellent add-on.  

Reviewer 3 Report

Aliotta et al have reviewed platelet function defects with a focus on procoagulant platelets. This is a very timely and well written review summarizing the field and recent advances. I only have some minor edits and suggestions

Specific Points

1) The generation of procoagulant activity is associated with membrane remodelling and the generation of PS positive procoagulant microvesicles which are important in providing a huge surface area for thrombin generation. These should be mentioned where relevant and included in the methods for measuring procoagulant activity.

2) Procoagulant platelets have sometimes been called zombie platelets due to their aggregation defects. This should be added to the list of names given in section 4.

3) The authors have omitted an important reference :- Histone H4 induces platelet ballooning and microparticle release during trauma hemorrhage.  Vulliamy P, Gillespie S, Armstrong PC, Allan HE, Warner TD, Brohi K.Vulliamy P, et al. Proc Natl Acad Sci U S A. 2019 Aug 27;116(35):17444-17449. doi: 10.1073/pnas.1904978116. Epub 2019

This needs including as its one of the few papers to describe procoagulant balloons in a clinical condition i.e. trauma and therefore contributing to the observed dysfunction of platelets in trauma

4) As PS platelets and vesicles significantly contribute to thrombin generation then various procoagulant assays should be mentioned in the table of methods including  ETP and  PPL for example?
